# Risk of Contralateral Breast Cancer in Women with and without Pathogenic Variants in *BRCA1, BRCA2*, and *TP53* Genes in Women with Very Early-Onset (<36 Years) Breast Cancer

**DOI:** 10.3390/cancers12020378

**Published:** 2020-02-07

**Authors:** Zerin Hyder, Elaine F. Harkness, Emma R. Woodward, Naomi L. Bowers, Marta Pereira, Andrew J. Wallace, Sacha J. Howell, Anthony Howell, Fiona Lalloo, William G. Newman, Miriam J. Smith, D Gareth Evans

**Affiliations:** 1Manchester Centre for Genomic Medicine, Manchester University NHS Foundation Trust, Manchester M13 9WL, UK; Zerin.Hyder@mft.nhs.uk (Z.H.); Emma.Woodward@mft.nhs.uk (E.R.W.); Naomi.Bowers@mft.nhs.uk (N.L.B.); marta.pereira@mft.nhs.uk (M.P.); Andrew.Wallace@mft.nhs.uk (A.J.W.); Fiona.Lalloo@mft.nhs.uk (F.L.); william.newman@manchester.ac.uk (W.G.N.); miriam.smith@manchester.ac.uk (M.J.S.); 2Division of Evolution and Genomic Sciences, School of Biological Sciences, Faculty of Biology, Medicine and Health, University of Manchester, Manchester Academic Health Science Centre, Manchester M13 9PL, UK; 3Division of Informatics, Imaging and Data Sciences, School of Health Sciences, Faculty of Biology, Medicine and Health, University of Manchester, Manchester M13 9PL, UK; Elaine.F.Harkness@manchester.ac.uk; 4Prevent Breast Cancer Centre, Wythenshawe Hospital, Manchester University NHS Foundation Trust, Wythenshawe, Manchester M23 9LT, UK; Sacha.Howell@christie.nhs.uk (S.J.H.); Anthony.Howell@manchester.ac.uk (A.H.); 5Manchester Breast Centre, The Christie NHS Foundation Trust, Wilmslow Road, Manchester M20 4BX, UK; 6Division of Cancer Sciences, Faculty of Biology, Medicine and Health, University of Manchester, Manchester Academic Health Science Centre, Manchester M13 9PL, UK

**Keywords:** breast cancer, contralateral, pathogenic variants, early-onset breast cancer, *BRCA1*, *BRCA2*, TP53

## Abstract

Early age at diagnosis of breast cancer is a known risk factor for hereditary predisposition and some studies show a high risk of contralateral breast cancer in *BRCA1* carriers diagnosed at very young ages. However, little is published on the risk of *TP53* carriers. 397 women with breast cancer diagnosed <36 years of age were obtained from three sources: (i) a population-based study of 283 women diagnosed sequentially from 1980–1997 in North-West England, (ii) referrals to the Genomic Medicine Department at St Mary’s Hospital from 1990–2018, and (iii) individuals from (i) and the Family History Clinic at Wythenshawe Hospital South Manchester who tested negative for pathogenic variants (PV) in all three genes. Sequencing of *BRCA1, BRCA2*, and *TP53* genes was carried out alongside tests for copy number for PV on all referred women. Rates of contralateral breast cancer were censored at death, last assessment, or risk-reducing mastectomy. In total, 47 *TP53*, 218 *BRCA1*, and 132 *BRCA2* PV carriers were identified with breast cancer diagnosed aged 35 years and under, as well as a representative sample of 261 not known to carry a PV in *BRCA1*, *BRCA2*, and *TP53*. Annual rates of contralateral breast cancer (and percentage of synchronous breast cancers) were *TP53*: 7.03% (4.3%), *BRCA1*: 3.57% (1.8%), and *BRCA2*: 2.63% (1.5%). In non-PV carriers, contralateral rates in isolated presumed/tested non-carrier cases with no family history were 0.56%, and for those with a family history, 0.69%. Contralateral breast cancer rates are substantial in *TP53, BRCA1*, and *BRCA2* PV carriers diagnosed with breast cancer aged 35 and under. Women need to be advised to help make informed decisions on contralateral mastectomy, guided by life expectancy from their index tumor.

## 1. Introduction

Lifetime risk of breast cancer in women with *BRCA1, BRCA2*, and *TP53* PVs ranges from 40–90% [1,2,3,4] with breast cancer at an early age of onset increased in women with a hereditary predisposition [5,6]. Contralateral breast cancer is also increased in *BRCA1* and *BRCA2* PV carriers [7]. A meta-analysis reported that 5- and 10-year cumulative risks of contralateral breast cancer were 15% and 27% respectively in *BRCA1* PV carriers [7]. Risks were also increased, but to a lesser extent in *BRCA2* carriers with 5- and 10-year risks of 9% and 19% respectively [7]. However the increase in risk contrasted notably to those without a *non-BRCA* PV in whom 5-year cumulative risk was estimated to be 3% increasing to 5% at 10 years [7]. The risk of contralateral breast cancer appeared to be greatest for women diagnosed at younger ages (<50 years) with 10-year cumulative risks ranging from 18.4%–23.9%, compared to 14.7% in women diagnosed aged 50 years or older [7]. We have previously reported that the risk of developing contralateral breast cancer is increased in *BRCA1* PV carriers diagnosed with breast cancer at a very young age (<30 years at diagnosis) [8]. In this study, of consecutive breast cancer cases with up to 20-years of follow-up, the annual rate of contralateral breast cancer overall was 0.6% but increased to 2%–3% per year among *BRCA1/2* PV carriers.

The risk of contralateral breast cancer for carriers of PVs in *TP53* is less well established. The population prevalence of *TP53* PV carriers is much lower, 1 in 5000 to 1 in 20,000 people [9], than *BRCA1/2* PVs which are estimated to occur among 1 in 300 to 1 in 800 individuals [10]. However, it is estimated that 5%–8% of hereditary breast cancers diagnosed under 30 years and without a *BRCA1/2* PV will have a *TP53* PV [9]. The cumulative incidence of breast cancer in women with *TP53* PVs is estimated to be 85% by the age of 60 years [11] and the risk of second cancers, associated with Li-Fraumeni syndrome (LFS), is high. Due to the low prevalence of *TP53* PVs, studies examining the risk of contralateral breast cancer have been limited.

In this study, we determine the risk of contralateral breast cancer in *BRCA1*, *BRCA2*, and *TP53* PV carriers compared to non-carriers in an extended series of *BRCA1/2* and *TP53* index cases diagnosed with breast cancer aged 35 years and under. We focus specifically on the risk of contralateral breast cancer in *TP53* PV carriers where existing research is limited.

## 2. Methods

Women diagnosed with breast cancer including ductal carcinoma in situ (CIS) 35 years and under were obtained from three sources (see Figure 1):A population-based study of 287 women aged 30 or under and diagnosed with breast cancer between January 1980 and December 1997 and recorded in the North-West of England cancer registry. Living participants provided consent for blood samples to be tested for *BRCA1, BRCA2,* and *TP53* PVs (*n* = 125; 39 PV carriers).women that were proven carriers (*n* = 358) from 854 *BRCA1*, 851 *BRCA2*, and 74 *TP53* positive families referred to the Genomic Medicine Department at St Mary’s Hospital, Manchester from 1990–2018, and not identified from 1.women without known PVs in the three genes were ascertained from (1) and the Family History Clinic at Wythenshawe Hospital South Manchester after testing negative for PVs in all three genes following a diagnosis of breast cancer aged 35 years and under (*n* = 14).

Breast cancer diagnoses were confirmed using hospital and pathology records. Family histories of malignancies were confirmed through the cancer registry, hospital notes, and death certificates. Family history was classified as positive if there were any first or second-degree relatives with breast cancer under 65 years of age, or ovarian cancer at any age, at the time of an individual’s breast cancer diagnosis.

Sanger sequencing of all exons of *BRCA1*, *BRCA2*, and *TP53* was performed until 2015, and thereafter by next generation sequencing. Initial screening in the study of women aged 30 or under was by Sanger sequencing; however, this has since been supplemented with next generation sequencing for an extended gene panel for those testing negative with two additional PVs identified.

Copy number variations were identified by Multiplex Ligation-dependent Probe Amplification (MLPA) and pathogenicity of variants were confirmed by the American College of Medical Genetics and Genomics (ACMG) standards and guidelines [12]. All index cases had testing of *BRCA1*/*BRCA2* and *TP53*. One woman with bilateral breast cancer was found on re-examination of her DNA to have a *BRCA2* PV in addition to her previously identified *BRCA1* PV. We are aware of 5 additional compound heterozygotes for *BRCA1* and *BRCA2,* but these are aged > 35 years.

Time to contralateral breast cancer was taken from date of initial breast cancer diagnosis. Individuals were censored at date of contralateral breast cancer, date of death, date of last assessment, or date of risk-reducing mastectomy, whichever was earliest. Synchronous bilateral breast cancer was defined as breast cancer diagnoses within 3 months of each other. Age at breast cancer diagnosis and time to diagnosis of contralateral breast cancer were compared using the Mann-Whitney test. Annual rates for contralateral cancer were calculated as the number of contralateral breast cancers as a proportion of the total years of follow-up until date of censor. Cumulative incidence curves for contralateral breast cancer were assessed using Kaplan–Meier curves and the log-rank test to compare differences between PV carrier status (*BRCA1, BRCA2*, and *TP53*). We also calculated left censored cumulative incidence curves for PV carriers where the date of the genetic report occurred after date of initial breast cancer diagnosis. *p*-values less than 0.05 were regarded as statistically significant. Analysis was conducted in Stata version 14.

Ethics approval for the study was through the North Manchester Research (08/H1006/77) and University of Manchester (08229) Ethics committees.

## 3. Results

In total, 398 women with breast cancer diagnosed aged 35 years and under were identified as PV carriers which included 47 *TP53*, 218 *BRCA1* and 132 *BRCA2* carriers (Figure 1, Table 1) and one individual with both a *BRCA1* and *BRCA2* PV. In addition, a sample of 261 women with breast cancer aged 35 years and under, with 100 shown not to carry a variant, was identified (Figure 1). 14 contralateral breast cancers were diagnosed in *TP53* PV carriers (2 synchronous), 54 in *BRCA1* carriers (4 synchronous), and 30 in *BRCA2* carriers (2 synchronous), compared to 13 contralateral breast cancers (0 synchronous) among non-carriers. The median age at initial breast cancer diagnosis for all PV carriers was 32.0 years (range 16.3 to 35.97) compared to 28.9 years (range 18.0 to 35.97) in non-carriers. This difference was statistically significant (*p* < 0.001). Compared to non-carriers (median age 28.9 years, range 18.0 to 35.97), the median age at initial breast cancer diagnosis was significantly higher for *BRCA1* (median 32.1 years, range 20.7 to 35.97, *p* < 0.001) and *BRCA2* carriers (median 32.9 years, range 22.4 to 35.9, *p* < 0.001) but not for *TP53* carriers (median 29.4 years, range 16.3 to 35.9, *p* = 0.224). In PV carriers, median time to diagnosis of contralateral breast cancer was approximately 7 years 7 months (ranging from 0 years to 29 years 6 months), while in non-carriers the median time to contralateral breast cancer was 9 years 9 months (range 1 year 7 months to 18 years 7 months) (*p* = 0.343). There were no statistically significant differences between median time to diagnosis of contralateral breast cancer for any of the individual PVs.

Annual rates of contralateral breast cancer (and proportion of synchronous breast cancers) were highest among *TP53* carriers at 7.03% (4.3%), followed by *BRCA1* carriers at 3.57% (1.8%) and *BRCA2* carriers at 2.63% (1.5%) (see Table 1). Table 2 depicts the annual rates of contralateral breast cancer when left censored at date of PV report. *BRCA1* and *BRCA2* PV carriers had lower annual rates of contralateral breast cancer (2.5% and 1.1%, respectively) compared to *TP53* carriers (10.7%). The proportion with synchronous breast cancer were 0.0% for *BRCA1* 1.6% for *BRCA2* and 5.3% for *TP53*.

Table 3 and Figure 2 show the cumulative risk of developing contralateral breast cancer for PV carriers (excluding synchronous breast cancers). The 10 and 20-year cumulative risk of contralateral breast cancer was 32.3% (95% CI 24.0–42.6) and 56.6% (95% CI 44.8–69.1) for *BRCA1*, 20.7 (95% CI 13.0–32.1) and 45.3% (95% CI 31.4–61.9) for *BRCA2* and 53.1% (29.6–80.6) and 82.4% (95% CI 49.9–98.8) for *TP53* carriers. The risk of contralateral breast cancer was significantly higher in *TP53* carriers compared to *BRCA1* (χ^2^ = 5.28, *p* = 0.022) and *BRCA2* carriers (χ^2^ = 11.9, *p* = 0.001). However, there was no significant difference between *BRCA1* and *BRCA2* carriers (χ^2^ = 2.00, *p* = 0.157). The cumulative risk of contralateral breast cancer by including only those women with PV report testing after the initial diagnosis of breast cancer is shown in Figure 3 and Table 3. The difference in cumulative risk between *TP53* carriers compared to *BRCA1/2* carriers remained statistically significant.

Of the 261 women diagnosed with breast cancer who were proven or presumed non-carriers, information on family history was available for just under half (127/261, 48.7%) including all 100 individuals who tested negative. In total, there were 13 contralateral breast cancers, of which 46% (6/13) occurred in those without a family history and 30.8% (4/13) in those with a family history. The remaining three contralateral breast cancers were in those with unknown family history status. The annual rates of contralateral breast cancer in all non-carriers, irrespective of family history, were much lower (0.44%), and there was little difference between women with and without a family history of breast cancer with annual rates of contralateral breast cancer of 0.69 and 0.56% respectively (Table 1). Figure 4 shows the cumulative risk of contralateral breast cancer development in non-carriers with and without a family history of breast cancer and demonstrates no significant difference between the groups (χ^2^ = 0.10, *p* = 0.757).

## 4. Discussion

Breast cancer remains a substantial health burden, not only in relation to the treatment of the primary tumor, but also the high risk of secondary cancers in inherited tumor syndromes. While contralateral breast cancer risk in *BRCA1* and *BRCA2* carriers has been well-described, there is a paucity of research on contralateral risk in LFS, owing to the low prevalence of germline *TP53* PVs in the general population. Our study is the first to refine the risks of contralateral breast cancer in this cohort, confirming substantial annual and cumulative risks in *TP53* carriers diagnosed with their first breast cancer aged 35 and under.

LFS is associated with an increased risk of several distinctive malignancies, with breast cancer being the most common cancer diagnosis between the ages of 18 and 44 years [11]. Accordingly, the risk of developing multiple primary cancers is considerable. A retrospective cohort study of 200 *TP53* PV carriers found a 30-year cumulative probability of second cancers of 57%, and similar to the findings in our cohort, 30% of individuals with multiple cancers were diagnosed with a second breast cancer (9/30) [13]. In the National Cancer Institute’s LFS Study, half of individuals developed a second primary cancer, with breast cancer being the most common second cancer diagnosis. Notably, the annual hazard/risk of developing a second cancer was similar to that of developing the first cancer, although the authors were unable to determine whether second cancers may be a result of treatment for the first cancer [11].

While there is no international consensus regarding screening for other types of cancers in LFS, risk-reducing mastectomy or annual MRI from the age of 20–49 years with avoidance of mammography is the current recommendation in the UK [14]. Regarding the management in *TP53* PV carriers, mastectomy, rather than breast-conserving surgery, is recommended for treatment of primary breast cancer. Treatment should aim to avoid radiotherapy or use with extreme caution, given the increased risk of radiation-induced tumors [15]. Given limited data on previous treatments, it is difficult to determine whether any of the contralateral cancer diagnoses in our LFS cohort were related to treatment of the primary breast cancer. Nevertheless, a 10-year cumulative risk of contralateral breast cancer of over 50% strengthens the recommendation for bilateral mastectomy after a primary breast cancer LFS [9]. There was little follow-up beyond 45 years of age and we cannot comment on risk in older women; however, in reality, the vast majority of *TP53* carriers develop breast cancer under 40 years of age.

Annual rates and cumulative risks of contralateral breast cancer in *BRCA1* and *BRCA2* PV carriers were significantly raised in our study compared to non-carriers, though to a lesser extent than *TP53* carriers. Cumulative risks were attenuated when analyzing only from the date of PV genetic report testing rather than initial diagnosis of breast cancer. This may indicate some bias in testing those with contralateral breast cancer. Malone et al. reported 10-year cumulative risks of contralateral breast cancer of 31.6% (*BRCA1* PV carriers) and 26.6% (*BRCA2* PV carriers) in those diagnosed between 30–34 years, depicting the higher contralateral risks in *BRCA1* PV carriers than *BRCA2* PV carriers, which is also demonstrated in our study [16]. A large multicenter study by Rhiem et al. (2012) additionally demonstrated similar 10-year cumulative risks for women diagnosed with their first breast cancer under the age of 40 years (30.1% 10-year risk for *BRCA1* carriers, 18.2% for *BRCA2* carriers) [17].

Interestingly, data presented by Rhiem et al. (2012) and by the National Cancer Institute’s Surveillance, Epidemiology and End Results (SEER) database, found that incidence of contralateral breast cancer after 10 years in familial non-*BRCA1/2* PV carriers was similar to sporadic breast cancer (contralateral breast cancer risk of 2.5% to 12.6% in 10 years depending on estrogen-receptor status and age of onset) [18]. Akin to our findings, there was little difference in the cumulative risk of contralateral breast cancer in non-carriers with and without a family history. While family history combined with early age of diagnosis of relatives is an important risk factor in development of first breast cancer, our study suggests that the risk of contralateral breast cancers may not be as substantial [19]. However, our numbers are relatively small, and the sporadic group could have undetected carriers in those not sequenced.

To reduce the incidence of a contralateral breast cancer, the most effective treatment at the time of a primary breast cancer in a *BRCA1* or *BRCA2* PV carrier is bilateral mastectomy [20]. However, bilateral mastectomy is a major, irreversible surgery requiring extensive pre-procedure counselling, considering both the risk of complications related to surgical removal of breasts and/or reconstruction, and the difference in a woman’s outward appearance, psychosocial and physical wellbeing [21]. With regards to other risk modifiers, conflicting evidence regarding reduction of risk with bilateral salpingo-oophorectomy exists, but recent evidence suggests it may benefit in estrogen-receptor (ER) positive primary breast cancer diagnosed prior to age 50 years [20]. Additional management options proven to decrease contralateral risk include endocrine therapies, such as the selective ER modulator, tamoxifen. Longer durations of tamoxifen therapy, particularly in the setting of an ER-positive first breast cancer, are associated with higher reductions of contralateral breast cancer risk, with a demonstrated preventative effect persisting even after treatment has ceased [22,23].

In our study, 8 of 397 *TP53, BRCA1* and *BRCA2* PV carriers (2%) had a synchronous breast cancer diagnosis compared to 0 synchronous cancers in 261 non-carriers, reflecting the increased risk of a secondary cancer within 3 months in association with inherited tumor syndromes. Currently, NICE clinical guidelines state that genetic testing should be offered to an individual with breast cancer if their combined *BRCA1* or *BRCA2* PV carrier probability is 10% or more, with fast-track genetic testing (within 4 weeks of diagnosis) currently not recommended outside the context of a clinical trial [14]. Although identification of PV carriers around the time of diagnosis in young-onset primary breast cancers may tailor management options, including the option of bilateral mastectomy, the individual should be allowed an informed choice for referral to genetic testing, adequate pre-test counselling, and be given a considerable period of reflection regarding such irreversible surgery which may not be ideal in the scenario of urgent cancer treatment.

The strengths of the current study are that DNA samples have been obtained for most participants. The dataset included a large set of *BRCA1/2* and *TP53* PV carriers with long term follow-up, and breast cancer diagnoses were confirmed through cancer registries, medical records, and/or death certificates. In addition, there were a relatively large proportion of contralateral breast cancers among PV carriers. However, the number in *TP53* PV carriers was limited, and there were incomplete pathology details for those patients diagnosed prior to 1990 as grade and receptor status were not reported reliably before this (grade and receptor status were reported in 69 and 58% of carriers respectively). We also had limited information on treatment and in particular, endocrine therapy which is known to decrease the risk of contralateral cancer and could account for the lower rates in *BRCA2* PV carriers who more frequently had ER-positive cancers.

## 5. Conclusions

Contralateral breast cancer rates are very high for *BRCA1/2* and *TP53* PV carriers who develop breast cancer at a very young age (<36 years). The cumulative risk increased with time from diagnosis of initial breast cancer. The risk was highest among *TP53* PV carriers. Women with PVs in these genes should be informed of their high risk of developing a contralateral breast cancer to help guide their future risk management, including the option of bilateral/contralateral mastectomy.

## Figures and Tables

**Figure 1 cancers-12-00378-f001:**
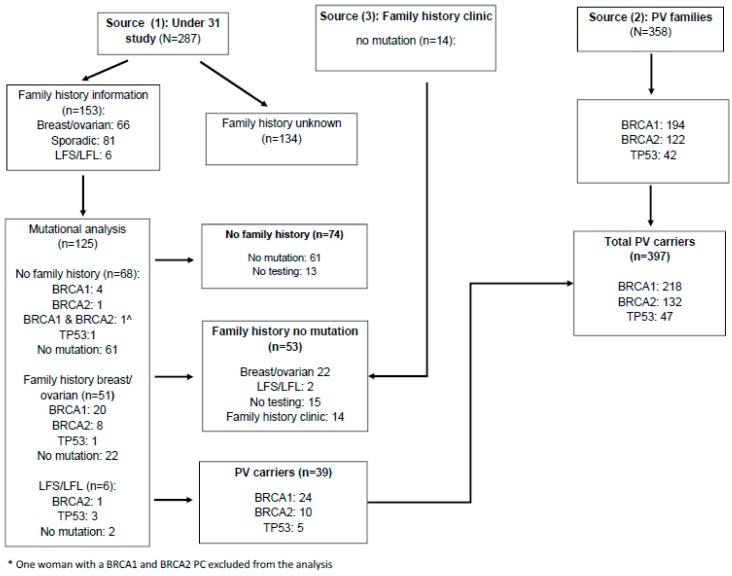
Flow chart showing data sources, family history, and carrier status.

**Figure 2 cancers-12-00378-f002:**
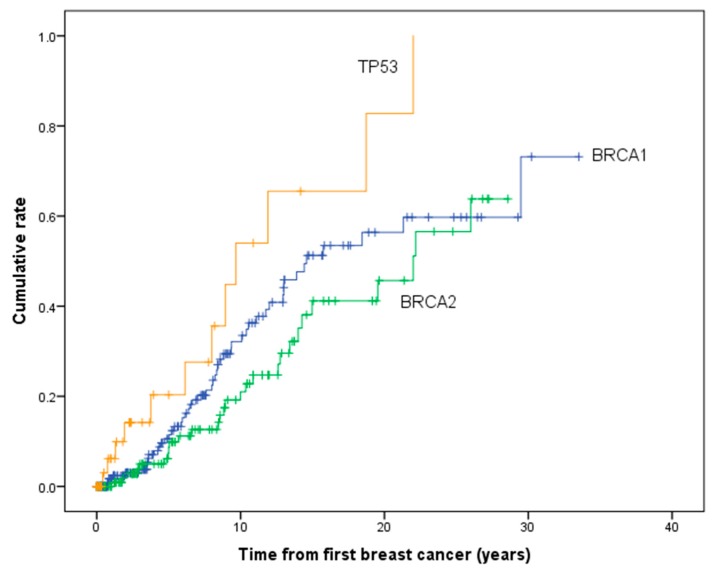
Cumulative risk of developing contralateral breast cancer in pathogenic variant carriers.

**Figure 3 cancers-12-00378-f003:**
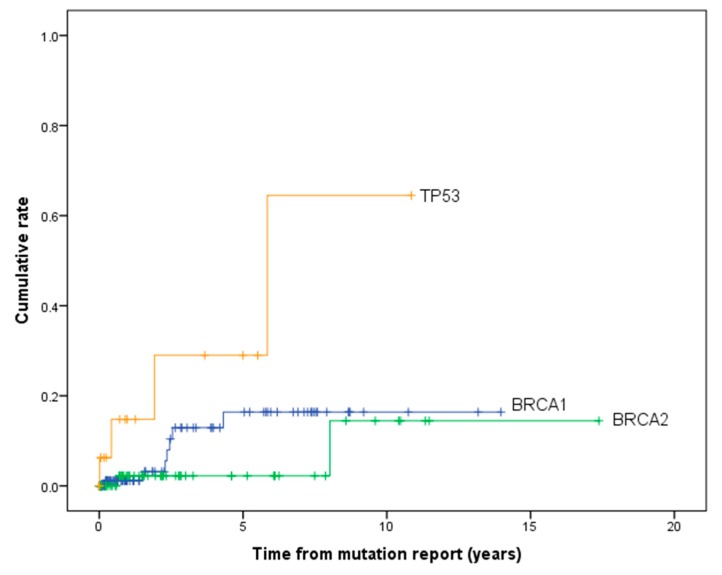
Cumulative risk of developing contralateral breast cancer in pathogenic variant carriers left censoring for date of PV report.

**Figure 4 cancers-12-00378-f004:**
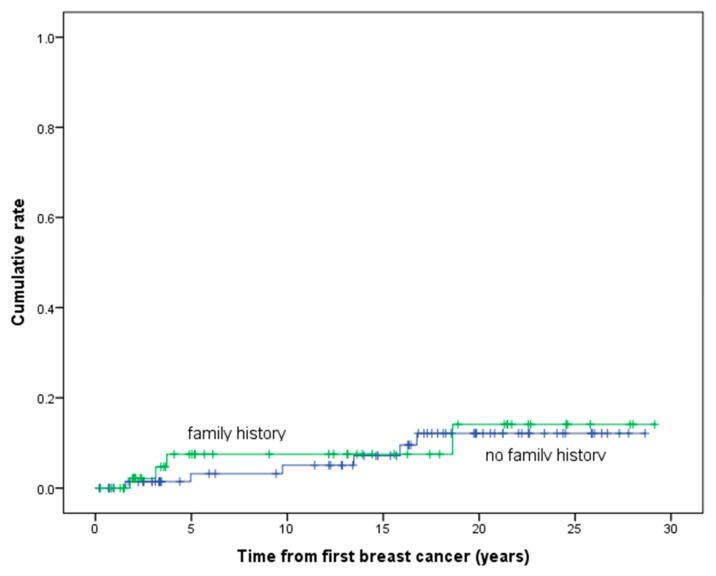
Cumulative risk of developing contralateral breast cancer in non-carriers by family history status.

**Table 1 cancers-12-00378-t001:** Number of pathogenic variant carriers and non-carriers with breast cancer <36 years.

Genetic Variant *	Total Breast Cancer (*n*)	Carrier Status	Contralateral Breast Cancer *n* (%)	Synchronous Breast Cancer *n* (%)	Risk-Reducing Mastectomy *n* (%)	Died *n* (%)	Person Years of Follow-Up	Annual Rate Contralateral Breast Cancer (Excluding Synchronous) (%)	Annual Rate All Contralateral and Synchronous Breast Cancer
*BRCA1*	218	Y	54 (24.8)	4 (1.8)	36 (16.5)	48 (22.0)	1403	3.57	3.85
*BRCA2*	132	Y	30 (22.7)	2 (1.5)	23 (17.4)	35 (26.5)	1063	2.63	2.82
*TP53*	47	Y	14 (29.8)	2 (4.3)	6 (12.8)	23 (48.9)	171	7.03	8.20
Total	397		98 (24.7)	8 (2.0)	65 (16.4)	106 (26.7)	2636	3.41	3.72
Family history	53	N	4 (7.5)	-		26 (49.1)	576	-	0.69
No family history	74	N	6 (8.1)	-		23 (31.1)	1074	-	0.56
Family history unknown	134	N	3 (2.2)	-		85 (63.4)	1312	-	0.23
Total	261		13 (5.0)	-		134 (51.3)	2963	-	0.44

* One individual had a BRCA1 and BRCA2 PV and is excluded from the table.

**Table 2 cancers-12-00378-t002:** Number of pathogenic variant carriers with breast cancer <36 years after left censoring at date of PV report.

Genetic Variant	Total Breast Cancers (*n*)	Contralateral Breast Cancers (*n*)	Synchronous Breast Cancers (*n*)	Years of Follow-Up	Annual Rate Contralateral Breast Cancer (Excluding Synchronous) (%)	Annual Rate All Contralateral and Synchronous Breast Cancers
*BRCA1*	108	7 (6.5)	0 (0.0)	280	2.50	2.50
*BRCA2*	63	3 (4.8)	1 (1.6)	187.0	1.07	1.60
*TP53*	19	5 (26.3)	1 (5.3)	37.6	10.65	13.31
Total	190	15 (7.9)	2 (1.1)	504.4	2.58	2.97

**Table 3 cancers-12-00378-t003:** Cumulative risk of developing contralateral breast cancer by time after initial breast cancer diagnosis (excludes synchronous).

Genetic Variant	All Women			Post PV Report		
	Years after Initial Breast Cancer	Cumulative Risk (%)	95% CI	Years after Initial Breast Cancer	Cumulative Risk (%)	95% CI
*BRCA1*	5	10.5	6.37–17.2	5	16.6	8.0–32.5
	10	32.3	24.0–42.6	10	16.6	8.0–32.5
	15	51.5	40.7–63.4	15	16.6	8.0–32.5
	20	56.6	44.8–69.1	20	16.6	8.0–32.5
*BRCA2*	5	7.2	3.5–14.6	5	2.0	0.3–13.5
	10	20.7	13.0–32.1	10	15.1	2.9–60.4
	15	40.9	28.3–56.4	15	15.1	2.9–60.4
	20	45.3	31.4–61.9	20	15.1	2.9–60.4
*TP53*	5	19.1	8.0–41.8	5	27.9	9.3–66.6
	10	53.1	29.6–80.6	10	56.8	20.5–95.3
	15	64.9	38.2–89.7	15	56.8	20.5–95.3
	20	82.4	49.9–98.8	20	56.8	20.5–95.3

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
