# Peer review of "Risk of Contralateral Breast Cancer in Women with and without Pathogenic Variants in BRCA1, BRCA2, and TP53 Genes in Women with Very Early-Onset (<36 Years) Breast Cancer"

_cancers, 2020, doi:10.3390/cancers12020378_

Round 1

Reviewer 1 Report

This paper covers an important topic: risk of contralateral breast cancer in women with BRCA1, BRCA2 and TP53 genes. However, there are quite a lot of grammar mistakes and reporting problems.

For example in line 145: "In total, there were 14 contralateral breast cancers, of which 9.3% (7/75) occurred 145 in those without a family history and 7.5% (4/53) in those with a family history." The two percentages should be 50% and 28.5%.

Also the first three sentences in the discussion.

Please add another column in Table 1 to indicate carrier and non-carrier.

...

Author Response

Reviewer 1

This paper covers an important topic: risk of contralateral breast cancer in women with BRCA1, BRCA2 and TP53 genes. However, there are quite a lot of grammar mistakes and reporting problems.

For example in line 145: "In total, there were 14 contralateral breast cancers, of which 9.3% (7/75) occurred 145 in those without a family history and 7.5% (4/53) in those with a family history." The two percentages should be 50% and 28.5%.

Thank you we have corrected this sentence as follows, line 168: “In total, there were 14 contralateral breast cancers, of which 50% (7/14) occurred in those without a family history and 28.6% (4/53) in those with a family history.” 

Also the first three sentences in the discussion.

Thank you we have deleted these sentences

Please add another column in Table 1 to indicate carrier and non-carrier.

Thank you this has now been added to Table 1.

Reviewer 2 Report

In the manuscript, the authors have tried to elucidate the effect of pathogenic mutations in BRCA1, BRCA2 and TP53 genes on the risk of contralateral breast cancer in women with early onset breast cancer.

Even though there had been several previous investigations dealing with nearly same topics as the authors’ title mostly using BRCA1/2 mutations, the authors’ manuscript is novel for dealing with additional TP53 pathogenic mutations.

First Comment,

Did the authors confirm the exclusivity or co-occurrence for BRCA1, BRCA2 and TP53 pathogenic mutations for each patient?

That is, did the authors consider the case of the co-occurrence in their calculations?

Second Comment,

The authors can use TCGA BRCA dataset (mutations and CNV data) to assess the results from their own dataset and to compare between them.

Third Comment,

In the 89 line, the authors should use word “classified”, instead of the word “classed” in the sentence.

Author Response

Reviewer 2

In the manuscript, the authors have tried to elucidate the effect of pathogenic mutations in BRCA1BRCA2 and TP53 genes on the risk of contralateral breast cancer in women with early onset breast cancer.

 Even though there had been several previous investigations dealing with nearly same topics as the authors’ title mostly using BRCA1/2 mutations, the authors’ manuscript is novel for dealing with additional TP53 pathogenic mutations.

Thankyou

 First Comment,

Did the authors confirm the exclusivity or co-occurrence for BRCA1BRCA2 and TP53 pathogenic mutations for each patient?

That is, did the authors consider the case of the co-occurrence in their calculations?

Thank you. We have added the following text to line 99:  “All index cases had testing of BRCA1/BRCA2 and TP53. There were no co-occurrences in this series. We are aware of 5 compound heterozygotes for BRCA1 and BRCA2 but these are aged >35 years.”

 Second Comment,

The authors can use TCGA BRCA dataset (mutations and CNV data) to assess the results from their own dataset and to compare between them.

We are not sure how to relate TCGA dataset to our study as this doesn’t specifically look at contralateral rates, more the spectrum of genetic mutations across different types of cancer. The CNV rate of 16% in BRCA1 and 9% in BRCA2 is not significantly different to the overall 20% rate for index breast cancers in BRCA1 and 6% in BRCA2 in our region.

Third Comment,

In the 89 line, the authors should use word “classified”, instead of the word “classed” in the sentence.

Thank you we have corrected this on line 89.

Reviewer 3 Report

Dear Authors,

The presented research article is clinically relevant and addresses the very urgent need for early detection of contralateral breast cancer is BRCA1/2 and TP53 predisposed population. Overall the manuscript is well written. There are few spell check that can be corrected. Besides that your experimental design,analysis, interpretation and conclusion is well elaborated. you have also addressed the limitation of study.

Thank you.

Author Response

Reviewer 3

Dear Authors,

The presented research article is clinically relevant and addresses the very urgent need for early detection of contralateral breast cancer is BRCA1/2 and TP53 predisposed population. Overall the manuscript is well written. There are few spell check that can be corrected. Besides that your experimental design,analysis, interpretation and conclusion is well elaborated. you have also addressed the limitation of study.

Thank you.

Reviewer 4 Report

Risk of contralateral breast cancer in women with and without pathogenic variants in BRCA1, BRCA2 and TP53 genes in women with very early onset (<36 years) breast cancer.

Introduction

The focus of the manuscript once you get to the discussion is on the risk of contralateral breast cancer in women with TP53 mutations. The introduction does not reflect this. There is a cursory mention of LF syndrome at the end of the introduction.

Line 65: “….1 in 300 and 1 and 800 of the population” – reword

Methods

“….including carcinoma in situ” – presumably you mean only DCIS and not LCIS? Be specific.

A flow chart/figure showing the different sources of cases as well as those with available data (i.e., the number of women who were sequenced, who of the non-carriers had genetic data or family history etc)

For the women from source 2 – these were women from BRCA1/BRCA2/TP53 families. Were they themselves sequenced?

What proportion were sequenced using Sanger sequencing and what proportion were done sing NGS?

Results

Paragraph 1 – what are these p-values from?

Need to comment on how carriers had an older age at first dx and a shorter latency (time between first and second dx). Did the latency differ between carriers and non-carriers by what gene they had a mutation in? If you excluded synchronous cancers was there still a significant difference?

Lines 145-146: “In total, there were 14 contralateral breast cancers of which 9.3% (7/75) occurred in those without a family history and 7.5% (4/53) in those with a family history.”

These numbers are unclear – please clarify. I’m not sure what the % refer to and they do not total 14.

Discussion

Delete instructions from first paragraph.

Line 181: “In terms of breast cancer management, treatment should aim to avoid radiotherapy or use with extreme caution, given its association with increased risk of radiation-induced tumours.” – reword, it is not clear that “its” is referring to LFS.

Line 182: “Given limited data on previous treatments….”

This is the first mention of treatment. Who did you have treatment data on? Can you do any analysis taking treatment into account?

You state that this means that it is hard to tell if any of the cancers diagnosed in the LFS group were treatment related – however, chemo decreases risk of asynchronous contralateral breast cancer, as does tamoxifen. Radiation is associated with some increased risk – but presumably would not be used in someone with a TP53 mutation. Overall the issue of treatment is not discussed enough and is a major limitation or the study.

Line 194 “….carriers that are also evidenced in our cohort” – this is unclear reword

Line 198 “….this cohort” what cohort???

Need to put the family history result in better context with literature. It is surprising to me that FH Is not associated with CBC in this study.

Last paragraph on page 6 discusses risk reduction - these seem to all be referring to 1st primary breast cancer. Need to put in context of CBC.

Page 7, paragraph 1 – discuss guidelines and treatment management but only in the context of BRCA1/2 – need to extend this to include a discussion of TP53 mutation carriers

Last paragraph – from how this study is described it does not appear to population-based.

Tables

Change “years of follow-up” to “person years of follow-up”

Clarify that those people included in Table 2 are women who had genetic testing after a first breast cancer diagnosis? Is that right?

Author Response

Reviewer 4

Introduction

The focus of the manuscript once you get to the discussion is on the risk of contralateral breast cancer in women with TP53 mutations. The introduction does not reflect this. There is a cursory mention of LF syndrome at the end of the introduction.

-Thank you we have made our aims clearer at the end of the introduction as follows on page 2, line 72: “We focus specifically on the risk of contralateral breast cancer in TP53 PV carriers where existing research is limited.”

Line 65: “….1 in 300 and 1 and 800 of the population” – reword

-We have reworded as follows on line 64: “…amongst 1 in 300 to 1 in 800 individuals”

Methods

“….including carcinoma in situ” – presumably you mean only DCIS and not LCIS? Be specific.

- Yes we have just included DCIS and have made this explicit on line 75.

A flow chart/figure showing the different sources of cases as well as those with available data (i.e., the number of women who were sequenced, who of the non-carriers had genetic data or family history etc)

- Thank you, we have added a Flow chart as Figure 1 as suggested.

For the women from source 2 – these were women from BRCA1/BRCA2/TP53 families. Were they themselves sequenced?

Yes all included women from each gene category were proven carriers. We have clarified this in the text, line 81 as follows: women that were proven carriers (n=356)….

What proportion were sequenced using Sanger sequencing and what proportion were done using NGS?

Initial screening in the under 31 study was by Sanger sequencing however this has since been supplemented with NGS for an extended gene panel for those testing negative - two additional pathogenic variants were identified.  This has been added to the text as follows on line 93: “Initial screening in the under 31 study was by Sanger sequencing; however, this has since been supplemented with next generation sequencing for an extended gene panel for those testing negative with two additional pathogenic variants identified.”

Results

Paragraph 1 – what are these p-values from?

These p-values are from the Wilcoxon sign ranked test to compare differences in median age and median time to diagnosis of contralateral breast cancer. We have clarified this in the methods section as follows on line 104: “Age at breast cancer diagnosis and time to diagnosis of contralateral breast cancer diagnosis were compared using the Mann-Whitney test.”

Need to comment on how carriers had an older age at first dx and a shorter latency (time between first and second dx). Did the latency differ between carriers and non-carriers by what gene they had a mutation in? If you excluded synchronous cancers was there still a significant difference?

We have changed the text in the Results as follows, line 122:  The median age at initial breast cancer diagnosis for all PV carriers was 32 years (range 16.3 to 35.97) compared to a median age of 28.9 years (range 18.0 to 35.97) in non-carriers. This difference was statistically significant (p<0.001). Compared to non-carriers (median age 28.5 years, range 18.0 to 35.97), the median age at initial breast cancer diagnosis was significantly higher for BRCA1 (median 32.1 years, range 20.7 to 35.97, p<0.001) and BRCA2 carriers (median 32.9 years, range 22.4 to 35.9, P<0.001) but not for TP53 carriers (median 29.4 years, range 16.3 to 35.9, p=0.224).

Due to the relatively small number of synchronous cancers the results remain the same after excluding synchronous cancers therefore we have not incorporated this into the text. 

Lines 145-146: “In total, there were 14 contralateral breast cancers of which 9.3% (7/75) occurred in those without a family history and 7.5% (4/53) in those with a family history.” These numbers are unclear – please clarify. I’m not sure what the % refer to and they do not total 14.

-Thank you we have corrected this sentence as follows, line 167: “In total, there were 14 contralateral breast cancers, of which 50% (7/14) occurred in those without a family history and 28.6% (4/53) in those with a family history.” 

Discussion

Delete instructions from first paragraph.

-Thank you we have deleted these sentences

Line 181: “In terms of breast cancer management, treatment should aim to avoid radiotherapy or use with extreme caution, given its association with increased risk of radiation-induced tumours.” – reword, it is not clear that “its” is referring to LFS.

-Thank you we have changed as suggested to clarify for TP53 as follows, line 199: “Regarding the management in TP53 PV carriers, mastectomy, rather than breast-conserving surgery, is recommended for treatment of primary breast cancer.  Treatment should aim to avoid radiotherapy or use with extreme caution, given the  increased risk of radiation-induced tumours [15].”

Line 182: “Given limited data on previous treatments….”

This is the first mention of treatment. Who did you have treatment data on? Can you do any analysis taking treatment into account?

- We only have limited data on the under 31 study which was obtained from a population register. As there was more extensive data on PV carriers in more recent times we do not feel that the limited data is sufficient to be presented as a sub-group analysis

You state that this means that it is hard to tell if any of the cancers diagnosed in the LFS group were treatment related – however, chemo decreases risk of asynchronous contralateral breast cancer, as does tamoxifen. Radiation is associated with some increased risk – but presumably would not be used in someone with a TP53 mutation. Overall the issue of treatment is not discussed enough and is a major limitation or the study.

Thank you. We understand this but for the population based under 31 study the data was from a cancer registry with limited data on treatment

Line 194 “….carriers that are also evidenced in our cohort” – this is unclear reword

-Thank you we have changed to clarify as follows, line 213: “…depicting the higher contralateral risks in BRCA1 PV carriers than BRCA2 PV carriers, which is also evidenced in our study.”

Line 198 “….this cohort” what cohort???

We have clarified as follows, line 218: “Interestingly, data presented by Rhiem et al. (2012) and by the National Cancer Institute’s Surveillance, Epidemiology and End Results (SEER) database…”

Need to put the family history result in better context with literature. It is surprising to me that FH Is not associated with CBC in this study.

Thank you. These are small numbers and the sporadic group could have undetected carriers in those not sequenced.  We have added the following text on line 226: “However, our numbers are relatively small and the sporadic group could have undetected carriers in those not sequenced.”

Last paragraph on page 6 discusses risk reduction - these seem to all be referring to 1st primary breast cancer. Need to put in context of CBC.

Thank you we have changed to make more relevant to CBC and have changed as follows, line 228: “To reduce the incidence of a contralateral breast cancer….” and on line 236: “Additional management options proven to decrease contralateral risk include endocrine therapies, such as the selective ER modulator, tamoxifen.”

Page 7, paragraph 1 – discuss guidelines and treatment management but only in the context of BRCA1/2 – need to extend this to include a discussion of TP53 mutation carriers

Thank you we have changed as suggested and added text as follows on line 199: Regarding the management in TP53 PV carriers, mastectomy, rather than breast-conserving surgery, is recommended for treatment of primary breast cancer.  Treatment should aim to avoid radiotherapy or use with extreme caution, given the  increased risk of radiation-induced tumours [15]. Given limited data on previous treatments, it is difficult to determine whether any of the contralateral cancer diagnoses in our LFS cohort were related to treatment of the primary breast cancer. Nevertheless, a 10-year cumulative risk of contralateral breast cancer of over 50% strengthens the recommendation for bilateral mastectomy after a primary breast cancer LFS [9].

Last paragraph – from how this study is described it does not appear to population-based

The under 31 population is from a population based cancer registry from 1980-1997.

Tables

Change “years of follow-up” to “person years of follow-up”

-Thank you we have changed as suggested

Clarify that those people included in Table 2 are women who had genetic testing after a first breast cancer diagnosis? Is that right?

-No this only includes follow up after mutation testing date. For a number the first breast cancer diagnosis was after genetic testing. All second breast cancers after mutation testing were excluded.

Round 2

Reviewer 4 Report

Refer to the new Figure 1 in the Methods section first (currently first mentioned in the results)

Line 93: Now refer to the study as the “under 31 study” in places. If this is the study name then capitalize. Also the objective says that this is women dx with breast at age 35 years or younger. On line 77 it says 30 or younger – please clarify. Based on the results it looks like the age 35 or younger is correct – which again does not seem to align with the “under 31 study”

Line 124: should be 28.9 years

Line 168: I still don’t think this is right.
I think it should be: “(7/14) occurred in those without a family history and 28.6% (4/14) in those with a family history.”

Line 86 – should this really be n=12?

Figure 1 is still not clear – maybe if you could relate it directly to the three sources described in the methods. However, the numbers in the methods are also still not clear – in particular for source 2.

In response you mention that these cases all come from a population-based registry from 1980 – 1997. This is true for source 1 – but what about the others? These other sources seem to be from a clinic/hospital.

Author Response

Refer to the new Figure 1 in the Methods section first (currently first mentioned in the results)

Thank you we have added “(see Figure 1)” to the Methods section on line 76

Line 93: Now refer to the study as the “under 31 study” in places. If this is the study name then capitalize. Also the objective says that this is women dx with breast at age 35 years or younger. On line 77 it says 30 or younger – please clarify. Based on the results it looks like the age 35 or younger is correct – which again does not seem to align with the “under 31 study”

Thank you we have tried to clarify this on line 93 by saying the ‘study of women aged 30 or under’ as in line 77.  We included additional women aged 31-35 identified as carriers or screened negative for the genes with a family history of breast cancer where we have follow up from diagnosis.

Line 124: should be 28.9 years

We have changed this to ‘compared to 28.9 years’ on line 124, and have changed line 123 to read 32.0 years for consistency.

Line 168: I still don’t think this is right. 
I think it should be: “(7/14) occurred in those without a family history and 28.6% (4/14) in those with a family history.”

Thank you for pointing this out, this has now been corrected on line 174 – please note numbers have changed slightly due to updated analysis.

Line 86 – should this really be n=12?

Yes this is correct – although now updated to 14 from additional analysis.

Figure 1 is still not clear – maybe if you could relate it directly to the three sources described in the methods. However, the numbers in the methods are also still not clear – in particular for source 2.

Thank you, we have tried to make the flow chart clearer by adding the three data sources.

In response you mention that these cases all come from a population-based registry from 1980 – 1997. This is true for source 1 – but what about the others? These other sources seem to be from a clinic/hospital.

Thank you we have removed the text ‘it is population based, and’ on line 262.
